# Research on safety supervision and management system of China railway based on association rule and DEMATEL

**Jia Liu[1], Yansheng Wang** [1]*, **Cunbao Deng[1], Zhixin Jin[1], Gaolei Wang[2,3], Chen Yang[2,3], Xiaoyu Li[2,3]**

**1** College of Safety and Emergency Management Engineering, Taiyuan University of Technology, Taiyuan Shanxi, China, **2** Railway Safety Research Center, China State Railway Group Co. Ltd., Beijing, China, **3** Railway Science &Technology Research & Development Center, China Academy of Railway Sciences Corporation Limited, Beijing, China

* wangyansheng@tyut.edu.cn

**Data Availability Statement:** A total of 409 accident reports were collected from publications published by MRSSD and the National Railway Administration of the People's Republic of China; 110 were from the National Railway Administration

## Abstract

Safety management is a key issue in the railroad industry that needs to be continuously focused on. And it is essential to study causes of accidents for preventing accidents. However, there is a limited academic discussion on the systematic study of organizations and accidents, as well as their safety-related interactions and accidents, as opposed to human-caused disasters. Thus, the model of China's railway safety supervision and management system by sorting out the existing organizations involved in management in China is established in this paper. Firstly, social forces and auxiliary enterprises are specifically added to the model. And then, the relationship between organizations and accidents, as well as the relationship between safety interactions among organizations and accidents are explored by analyzing 224 accident reports, which led to 4 principles for accident prevention. Finally, based on these principles, measures to secure organizational nodes, as well as measures to promote safe interactions among organizations are proposed. The results showed that: (1) China Railway node is not only the most critical node in the safety supervision and management system but also the most vulnerable to the influence of other nodes. (2) The accident occurred due to the simultaneous occurrence of an accident at the China Railway node and the social force node. (3) When there are often safety risks in auxiliary enterprises and social forces simultaneously, the government's management is likely to be defective. The findings in this study can provide helpful references not only for improvement of safety management system structure and supervision and management mechanism but also for the formulation of safety supervision and management policies in China and other countries.

## 1 Introduction

Rail is becoming a more critical mode of transportation in China which promotes the development of raising the standard of living of the people [1,2]. However, with the continuous development and construction of railway transportation, railway accidents occur frequently and

of the People's Republic of China (https://www.nra.gov.cn) and 299 were from the "2009 Railway Traffic Accident Cases" (ISBN:9787113120993).

**Funding:** The research described in this paper was financially supported by National Natural science Foundation of China (52004175) and Science and technology innovation project of colleges and universities in Shanxi Province (2020L0105) from Yansheng Wang.

**Competing interests:** The authors have declared that no competing interests exist.

bring serious consequences, including casualties, property damage, and poor social impact, which greatly threatens the further development of the railway industry [3]. The guarantee of railway safety depends on the progress of related technology, the improvement of organization and system, and effective safety management [4,5]. For the rail, academic and practical attention has focused mostly on the role of technical and human-caused disasters such as improper structural design of rolling stock and misfeasance of human [6–8]. For example, the Wenzhou train crash accident on July 23, 2011, and with serious consequences. The accident was caused by an equipment defect [9]. However, accidents have not been caused by a coincidence of independent failures such as defect of equipment, but by a systematic migration of organizational behavior toward accident, which is described by Rasmussen and Baysari et al. in their article [10,11]. Previous studies show that 30%-40% of accidents can be attributed to organizational factors and almost all accidents were related to at least one organizational factor in large-scale complex systems. And the reasons are not only related to individual's fault but related to a partial or total failure of the organization [10,12,13]. The interaction problem of the organization may be another reason that affects safety [14,15]. Synergies among organizations that emerge from frequent interaction and communication between organizations are considered to be the basis for achieving system functionality [16]. Deficiencies in reliability of synergy among organizations lead to unexpected situations where risks are increase and rail can't be supervised and managed as administrator anticipated [17]. In summary, poor organization structure and insufficient interaction among the organizations can be a significant cause of accidents, which are also related to the management and regulatory [18,19].

It is obvious that the government regulation is an inadequate way to enforce safety. Safety-related coordination between the operator and the manufacture, as well as the coordination between media and regulators are considered to be effective ways to promote safety management [20,21]. Thus, organizations should take on the responsibility to respond to risks [22]. Organizations must adapt their structure to follow the changing safety objectives [23]. In China, the organizational structure of the railway industry changed in 2013 to a government-regulated, company-operated situation that continues to this day [24,25]. Within this organizational structure, the operator-regulator relationship has been identified as crucial causal factor for accident. Meanwhile, the information flow between corporations and the government can be an indication of overall safety [26,27]. The accident and mortality rate have dropped benefits from change of China. However, a number of accidents caused by inadequate organizations in recent years have drawn the attention of safety personnel. For example, trains T179 and D2809 derailed respectively on March 2020 and June 2022. Moreover, the Baiyin derailment accident that caused serious consequences happened on March 8, 2022 [28,29]. The accident highlighted the need for further improvement in the current organizational framework of railway safety management. Rail safety is affected by organizational behavior in a variety of ways [30]. Accident analysis methods, such as Model of Socio-Technical [31] and STAMP [32,33], are seen as an effective tool to help to research the impact of organizational behavior. Such as Xing [34] provided an administrative structure of accident by using STAMP, and some relationships between organizations and accidents are drawn. However, as this conclusion is based on the analysis of one accident and the complexity of the analysis method, there is a doubt that the conclusion apply to all accidents [35]. Current research elucidates that the process of accidents involves the entire safety management system and highlights the significance of organizations and safety-related interactions among the organizations in this process. Additionally, it reflects the inadequacy of current research: There is a lack of systematic research on accident and organization, as well as accident and safety-related interactions among organizations, especially in the domain of railway industry [36].

Therefore, establishing a safety supervision and management system which applies to China railway and investigating the mechanisms of safety-related interactions among organizations lead to the reduction of accidents and poor impact in the future. In view of this, we established safety supervision and management system model of China railway and then used association rule and DEMATEL method to research safety-related interactions relationships between organizations in the model [27,37,38]. Finally, based on the research results, we proposed some specific measures to strengthen the safety of railway management. The above issue has provided a reference for the safe operation of the railway and motivated researchers and engineers in the safety fields of transport to study more reasonable supervision and management operation mechanisms in the railway industry.

## 2 Theoretical model of China railway safety supervision and management

By summarizing and analyzing the current status of the primary responsibilities, divisions, and cooperation of the railway safety supervision and management department in China, a theoretical model of China's railway safety supervision and management is built. The theoretical model is consisted of three levels of nodes. The first-level organization nodes include government, China Railway, social forces, and auxiliary enterprises, the second-level organization nodes include government and China Railway subordinate units, news media, social organizations, and manufacturing enterprises, etc. Each secondary node is also subdivided into tertiary nodes. The theoretical model is shown in Fig 1. The organization nodes at the same level are

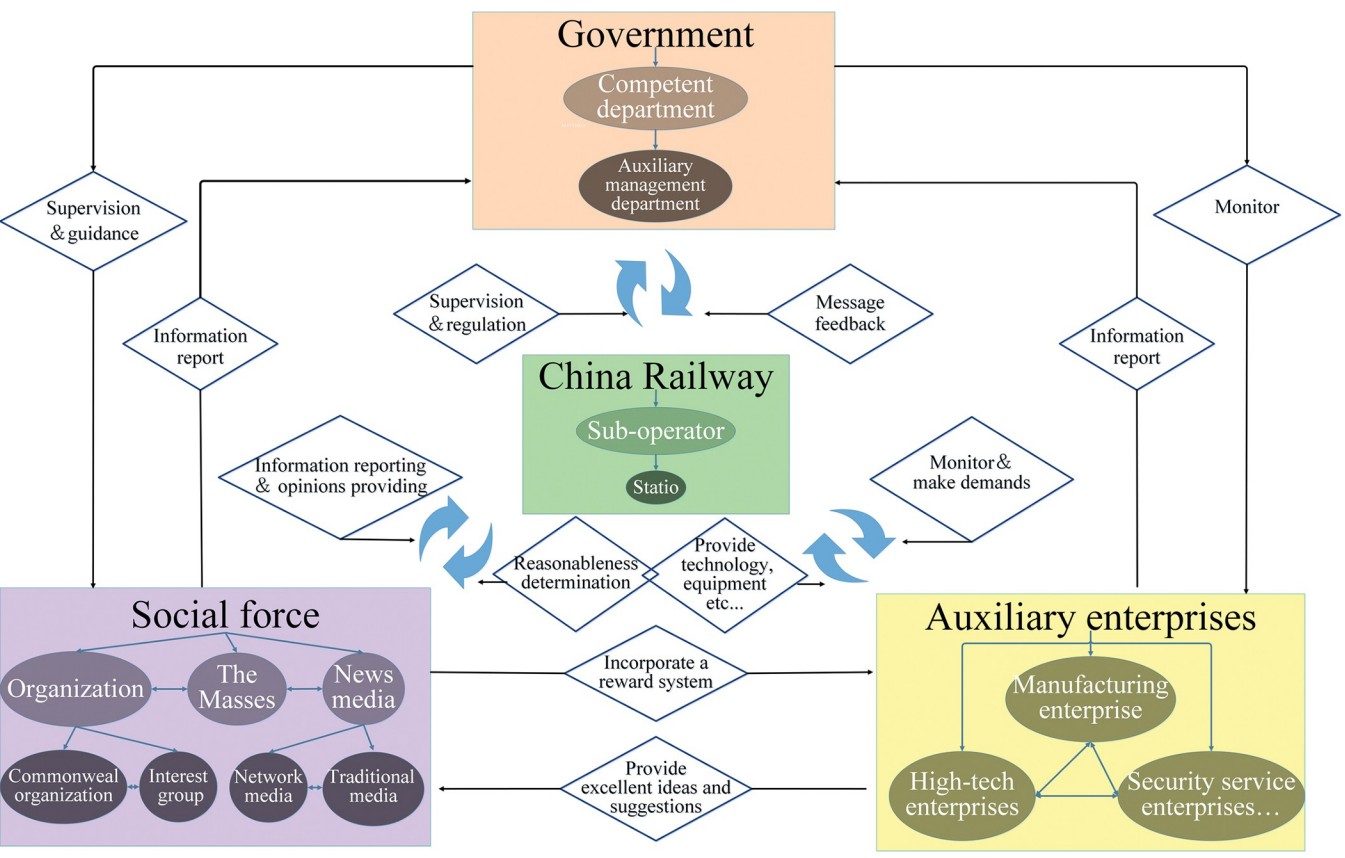

**Fig 1. Safety supervision and management system of China railway.**

connected to each other by chains that represent the inter-connection and interaction between nodes at the same level, in which arrows indicate the direction of the connection and interaction.

## 2.1 Nodes analysis

The government node is dominated by the national competent department. It is the direct leader in the China railway safety supervision and management system, the setter of safety objectives, the organizer and coordinator of handling accident, and the developer of relevant policies. Railway safety in China is directly managed by the railway officials at the government node, who also define safety goals and relevant regulations [39].

China Railway is the primary railway operator under the supervision and management of government departments and is also the safety goals setter. It is responsible for the scheduling and commanding of the railway, monitoring, early warning, risk management, etc., which is a power conferred by law and regulation.

Social forces are the demanders, guides, and supervisors of the safety. The China Railway often traverses a variety of complex terrains and has long routes and wide spans, which gives rise to the hidden dangers of natural disasters, difficult supervision, the lack of manpower, and the failure of full-time supervision in the whole line. Based on the above reasons, safety management is in urgent need of comprehensive participation of social forces.

Auxiliary enterprises are providers of technologies and services. The production and cooperation between manufacturing, safety service and high-tech enterprises are important guarantees in the continuous promotion of the construction of China railway safety [40,41]. China Railway take advantage of science and technology to improve disaster resilience and safety through the assistance of auxiliary enterprises.

## 2.2 Chains analysis

1. Government and China Railway
   The government is the maker of laws, regulations, systems, policies and requirements. These elements influence the goals, strategies and methods of railroad safety management, which in turn affects to some extent the quality of safety operations on China's railroads. China Railway, the main body of railway operations, is in charge of the operation of railway lines. It is also required to regularly report the safety management to the government department, and report in writing in a timely manner when hidden dangers or accidents are discovered.

2. Government and social forces
   The government can protect the reasonable rights of social forces when they participate in the railway safety management and restricts them through laws and policies.
   The hidden dangers and accident information can be discovered by social forces which can make up for the lack of government mobility. This is due to the characteristics of social forces, including wide and efficient participation, fewer constraints, diversified organization forms, flexible management mechanisms, and varieties of feedback channels.

3. Government and auxiliary enterprises
   The government mainly plays a leader and a guide in the development of auxiliary enterprises. As a leader, the government states mandatory regulations to ensure that auxiliary enterprises manufacture safely under the framework. The government should be market-oriented and introduce measures to assist the development of auxiliary enterprises while familiarizing itself with and grasping the laws of market operation. In particular, it should

help enterprises with strong market capacity, small size and low visibility to develop in a sustainable and stable manner.

Auxiliary enterprises could actively respond to government policies and cultivate innovative capabilities that meet the needs of the safe development of the China railway. They also accelerate the construction of modern enterprises and improve the quality and efficiency of products. In addition, they can help the government assume social responsibilities and relieve social pressure.

4. China Railway and social forces

The role of China Railway in influencing social forces is manifested in 3 main ways.

   a. China Railway acts as a manager to avoid disorder and self-governance, and it is responsible for coordinating and communicating with social forces. b. China Railway helps the internal coordination and resource integration of social forces. c. While in a disaster, China Railway can help match participants, promoting the efficiency of social forces.

Social forces use autonomy and creativity to help the China Railway find deficiencies in a traceable way through the supervision of the China Railway and seek solutions in a further step.

5. China Railway and auxiliary enterprises

The production and cooperation between enterprises are premises of sustainable and steady development and the maximization of the country's interests. China Railway s lessens the strain by purchasing ancillary businesses for cutting-edge safety goods and services. From auxiliary enterprises. Auxiliary enterprises profit from provision of China railway safety products and services. Auxiliary enterprises gain an advantage over the competition by pushing the boundaries of innovation, which contributes to developing railway safety products positively.

6. Social forces and auxiliary enterprises

Social forces transport talents and inject vitality into the long-term development of auxiliary enterprises. In addition, positive reporting helps to build up the image of the enterprise, which allows auxiliary enterprises to raise funds in society for development. Moreover, the increasing attention of auxiliary enterprises has also made them more active in participating in the safe construction of China railway.

Auxiliary enterprises provide social forces with an integration mechanism that rewards participation in China railway safety management. Auxiliary enterprises can adopt excellent ideas and suggestions from social forces. And they can absorb talents in an all-round way. Eventually, the enthusiasm of social forces is enhanced, while in the degree of participation on China's railway safety management and cultivates the spiritual strength of society.

## 3. Materials and methods

### 3.1 Data preparation

We collected a total of 409 accident reports from publications published by MRSSD and the National Railway Administration of the People's Republic of China, of which 110 were from the National Railway Administration of the People's Republic of China, and 299 were from the "2009 Railway Traffic Accident Cases". The above accident reports record the accident information in detail. According to the railway accident report, the pattern of occurrence of railway accidents is divided into three categories.

1. The factors that lead to the accident are independent of each other (Fig 2A)

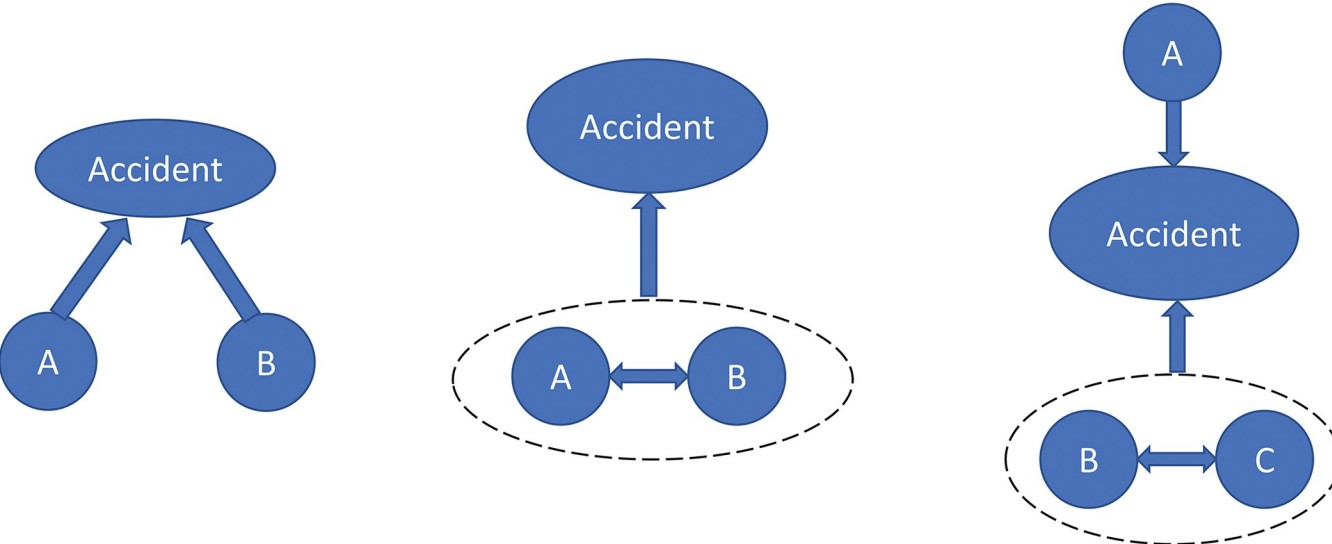

**Fig 2. The accident occurrence type.**

2. There is an interaction between factors (Fig 2B)

3. The factors are both independent and mutually influencing (Fig 2C)

This study focuses on railway accidents caused by the interaction between nodes (Fig 2B and 2C). According to the characteristics of the first-level nodes, the article divides the influence relationship between the first-level nodes into three categories. The three categories are as follows: (1) The $X_i$ node does not adequately influence the safety of the $X_j$ node; (2) The error of the $X_i$ node directly leads to the error of the $X_j$ node; (3) $X_i$ node has an optimized effect on the function of $X_j$ node.

Based on the above analysis, 224 valid railway accident reports were obtained after screening out the accident reports containing only one type of accident mode. The statistical rules are shown in Table 1. Frequency counts in Table 2 are obtained from the influence relationship between the first-level nodes, which are shown in Fig 2. The first-level nodes are consisted of Government $X_1$, China Railway $X_2$, Social force $X_3$, and Auxiliary enterprises $X_4$. If the description in the accident report conforms to the rules, the influence relationship of the nodes in the accident report is recorded as $< X_i, X_j >$, and the $< X_i, X_j >$:1 is recorded in Table 2. Similarly, if the cause of the accident in accident n of the accident report is that the government's behavior influenced the China Railway, n = $< X_1, X_2 >$: 1 was recorded. According to this rule, Table 2 is made by arranging 224 railway accident reports.

### 3.2 Association rule and DEMATEL method

After the determination of the organizations that make up the railway safety supervision and management system, there is a need to adopt the method in order to resolve the relationship between organizations in the model.

The primary purpose of this study is to determine the importance of shown organizations in Fig 1 and to analyze of safety-related interactions between them. To that end, The Decision-making Trial and Evaluation Laboratory Association Rules and DEMATEL method that includes a data mining technique and an effective weighting should be used.

**Table 1. Description of influence indicator.**

| Rank | Influence relation | Description |
|---|---|---|
| 1 | $<X_1, X_2>$ | Although the latter violates government regulations, mistakes can be discovered and stopped if the government strengthens inspection and supervision. |
| 2 | $<X_1, X_3>$ | The failure of the supervision of The Ministry of Public Security and other offices influences the latter's dangerous behavior; Relevant government supervision is not strict, and problems such as overloading, mixing people and goods, and driving without licenses have arisen. |
| 3 | $<X_1, X_4>$ | Illegal operations persist for a long time without detection and management. |
| 4 | $<X_2, X_1>$ | Promoting policy replenishment; Accelerate the renovation of grade crossings. |
| 5 | $<X_2, X_3>$ | Lack of communication mechanisms; Failure to properly use social forces to participate in rescue; Take the lead in establishing a long-term safety management mechanism to cover the shortage. |
| 6 | $<X_2, X_4>$ | The railway bureau does not manage the company's operation in place; Defaulting to unsafe behavior for accomplishment; Lack of strict control, including poor supervision and lax acceptance. |
| 7 | $<X_3, X_1>$ | Failed attention, such as how to deal with influences of psychopath. |
| 8 | $<X_3, X_2> <X_3, X_4>$ | Outsourcing workers, individual companies, and unrelated social workers have influences and refuse to obey management leading to uncontrolled behavior. |
| 9 | $<X_4, X_1> <X_4, X_2>$ | Concealment; Failure to disclose details or make troubles; Violations and mistakes in operation; Inadequate supervision of regulatory companies; Actions without application or consent; The quality of the equipment; Failure to comply with regulations having an influence on driving; Hire unqualified companies or people; Inadequate safety has an influence on driving; Unreported cases. |
| 10 | $<X_4, X_3>$ | Inadequate training and unqualified examination towards outsourced workers. |

**3.2.1 Association rule method.** Association rule is one of data mining technique that is often applied in areas such as risk management to investigate patterns of associations structures among sets of items [42,43]. The utilization of association rules is commonly employed to uncover significant and concealed correlations between attributes. The Apriori algorithm is the basic algorithm for mining item sets, and following from the development of the Apriori algorithm, the association rule technique becomes highly efficient. A standard association rule is of the form X→Y, where X is the antecedent and Y is the consequent. It is important to note that association rules solely establish associations between item sets and do not deduce direct causality.

In general, association rules can be summarised in the following two steps:

Step1: Identify frequent item sets

Support and Confidence are important parameters in the process of identifying frequent item sets. Support represents the probability of emergence of a rule or item set appearing in all transactions. Confidence is used in association rules to determine the frequency that Y occurs in transactions containing X. The greater the level of Confidence, the more likely the events that Y is included in the X will occur. The formula for Support and Confidence are in Eqs (1)

**Table 2. Frequency table of first level nodes influence matrix.**

|  | $X_1$ | $X_2$ | $X_3$ | $X_4$ | SUM |
|---|---|---|---|---|---|
| $X_1$ | 0 | 38 | 14 | 35 | 87 |
| $X_2$ | 8 | 0 | 12 | 23 | 43 |
| $X_3$ | 4 | 66 | 0 | 8 | 78 |
| $X_4$ | 7 | 112 | 8 | 0 | 127 |
| SUM | 19 | 216 | 34 | 66 | 335 |

 

and (2).

$$\text{Support}(X) = P(X) = \frac{\text{number of occurrences}(X)}{\text{total number of transactions}(T)} \qquad (1)$$

$$\text{Confidence}(X \rightarrow Y) = P(Y/X) = \frac{P(XY)}{P(X)} = \frac{\text{Support}(X \cup Y)}{\text{Support}(X)} \qquad (2)$$

Step2: Generating association rules from frequent item sets

By definition, these rules must satisfy min(sup) and min(conf), and lift>1. Association rule introduce Lift to compensate for the shortcomings that rules calculated only by confidence may conflict with the actual rules. Lift is calculated as shown in Eq (3).

$$\text{Lift}(X \rightarrow Y) = \frac{\text{Support}(X \cup Y)}{\text{Support}(X) \times \text{Support}(Y)} \qquad (3)$$

**3.2.2 DEMATEL method.** DEMATEL is considered an integrative approach to analyzing structural models with causal relationships, which analyses intertwined problems by delineating causal relationships [44,45]. The benefit of this approach is that it can visualize the interrelationship between factors through causal diagrams, which include direct and indirect influence relationships, making up for the lack of human extraction of accident reports that only consider the direct influence relationship [46]. Currently, this method has been widely used in numerous disciplines in the field of safety including emergency management [47], node importance assessment [48], identification of crucial accident causal factors [49] and so forth. Therefore, authors were motivated to employ DEMATEL in this study to explore safety-related interactions among the organizations. The steps of DEMATEL were then applied to this context as follows:

The basic steps of DEMATEL are shown below:

Step1: The statistics in Table 2 are used to indicate direct effect of the accident report each organization exerts i on every other organization j, denoted by using a scale that ranges from 0 to 9, which ranges step h = 13. These scores are used to create direct relations matrix M, which shows a pair-wise relationship of safety-related interactions among the organization. Direct relation matrix M is illustrated in Eq (4).

$$M = \begin{pmatrix} m_{11} & \cdots & m_{1j} \\ \vdots & \ddots & \vdots \\ m_{i1} & \cdots & m_{ij} \end{pmatrix} \qquad (4)$$

Step2: The direct relation matrix is normalized using a maximization method like Eq (5), which is considered a typical standardized method. All the values in the resulting matrix N are between 0 and 1.

$$N = \left( \frac{Z_{ij}}{|\max(Z_{ij})|} \right) i, j \in 1, 2, \dots, n \qquad (5)$$

Step3: From the normalized matrix N, total relation matrix T is obtained using Eq (6).

$$T = (N + N^2 + N^3 + \dots + N^k) = \sum_{k=1}^{\infty} N^k \qquad (6)$$

 

Step4: The rows and columns values of the matrix T are summed separately to obtain the matrices D and C, which represent the driving power and the dependency of the factors respectively. Eqs (7) and (8) are used as follows to calculate D and C:

$$D = (D_1, D_2, \ldots, D_n); D_i = \sum_{j-1}^{n} t_{ij}, (i = 1, 2, 3, \ldots, n) \tag{7}$$

$$C = (C_1, C_2, C_3, \ldots, C_n); C_i = \sum_{j-1}^{n} t_{ji}, (i = 1, 2, 3, \ldots, n) \tag{8}$$

Step5: Obtain the network relationship map by using Ri and Mi, which are calculate by (Di-Ci) and (Di+ Ci).

$R_i$, as horizontal axis value, represents the causal degree, and $M_i$, as the vertical axis value, represents the central degree in network relationship map. The small value of Di+ Ci implies that this part has limited impact on other parts, whereas a big value indicates that it is the pivotal driving force for resolving the critical part and should be prioritized.

## 4. Analysis and results

In this section, calculation and analysis results are shown using association rules and DEMATEL methods in order to explore the association rules and safety-related interactions among railway organizations of China.

### 4.1 Association rule mining of the relationship between nodes and accidents

The nodes that constitute safety supervision and management system of China railway are a basis for realizing railway infrastructure and operation functions. Taking the significance of nodes in the railway network into account, one of the critical causes of railway accidents is errors in the internal work of nodes. Therefore, the safety of controlling nodes is the primary task to ensure the effectiveness of railway safety management. According to the association rule analysis steps, the data of accident reports are analyzed as follows:

Step 1: Data preprocessing. Use "True" and "False" to describe the node state in the accident report, such as Eq (9).

$$X_k = \begin{cases} F & X_k \notin <X_i, X_j> \\ T & X_k \in <X_i, X_j> \end{cases} \tag{9}$$

step2: Considering the data structure, this research preset the minimum Support at 4%, the minimum confidence at 30%, and the Lift at Lift>1.

Step 3: Operation and rule selection. Association rules based on the Apriori algorithm were used in SPSS Modeler to guide the analysis.

Four rules are displayed in Table 3 after completing the above three steps.

During the calculation of the association rules, four strong associations and two medium associations were identified by calculating the frequency of occurrence of first-level nodes in 224 valid railway data as below. The correlation relationships between nodes are shown in Fig 3. The thickness of the connecting line in the figure indicates the possibility of nodes that appear simultaneously in one accident, and the greater probability is considered to be a strong association relationship between nodes, represented by purple line segments; Yellow line

**Table 3. Association rules analyze index values.**

| Rank | Consequent | Antecedent | Support (%) | Rule Support (%) | Confidence (%) | Lift |
|------|-----------|-----------|-------------|------------------|----------------|------|
| 1 | China Railway | Social force | 34.23 | 31.98 | 93.42 | 1.01 |
| 2 | Government | Social force<br>Auxiliary enterprises | 7.21 | 4.05 | 56.25 | 1.37 |
| 3 | Government | Social force<br>Auxiliary enterprises<br>China Railway | 4.95 | 2.70 | 54.55 | 1.33 |
| 4 | Social force | China Railway | 92.79 | 31.98 | 34.47 | 1.01 |

segments imply a moderate association relationship between nodes if the probability is low. Correlations are ranked as follows:

Strong association relationships: $(X_2, X_4) > (X_1, X_2) > (X_2, X_3) > (X_1, X_4)$
Moderate association relationships: $(X_1, X_3) > (X_3, X_4)$

## 4.2 Determination of the interrelationships between nodes

The influence relationship among nodes, in which the $X_i$ node directly or indirectly leads to the state change of the $X_j$ node. And the influence relationship plays a decisive role in the occurrence of railway accidents. The steps of DEMATEL were applied to study the safety-related interactions among the organizations nodes as follows:

Step1: Direct relationship matrix A

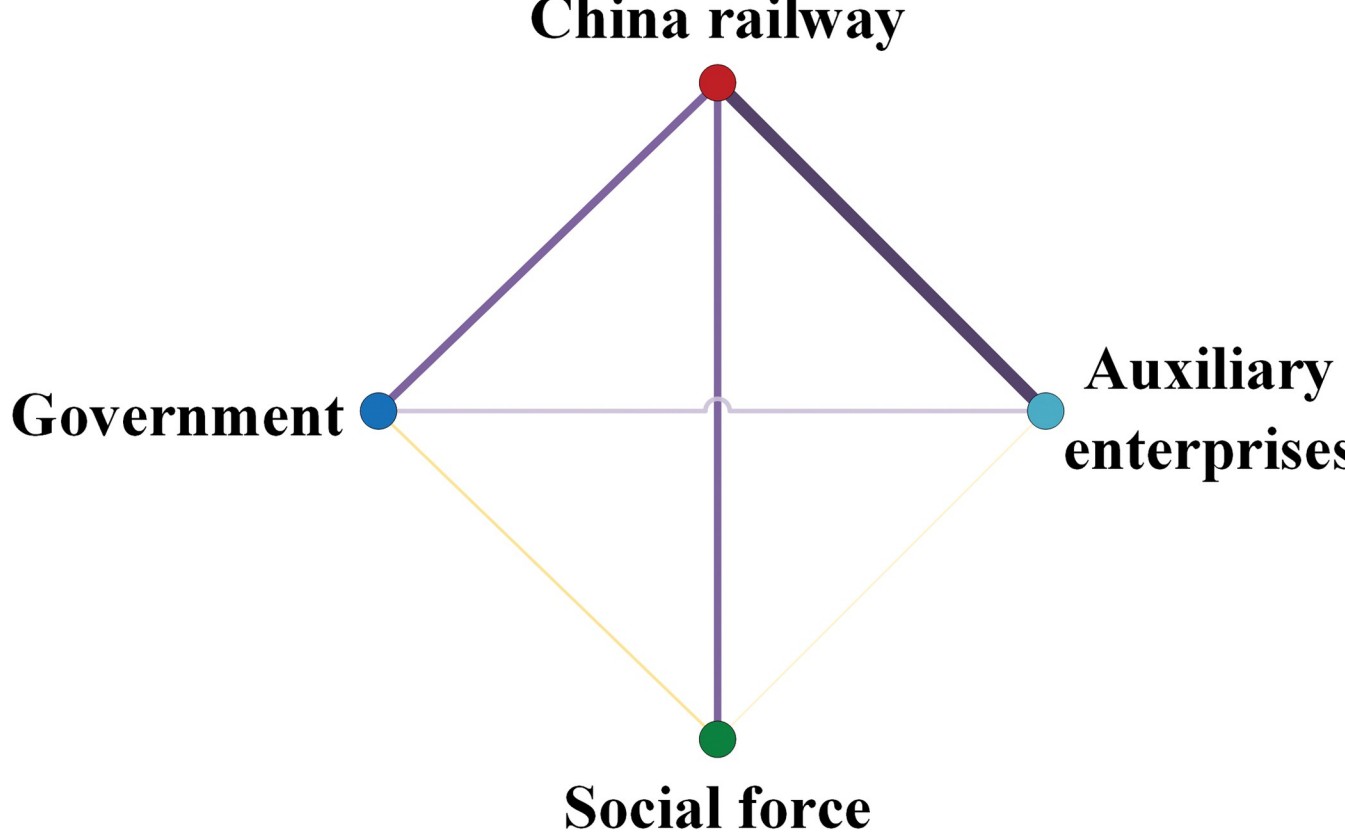

**Fig 3. The diagram of node association degree.**

**Table 4. Direct relation matrix.**

|  | Government | China Railway | Social force | Auxiliary enterprises |
|---|---|---|---|---|
| Government | 0 | 3 | 2 | 3 |
| China Railway | 1 | 0 | 1 | 2 |
| Social force | 1 | 6 | 0 | 1 |
| Auxiliary enterprises | 1 | 9 | 1 | 0 |

Following the DEMATEL steps described preceding part of the text, Step 1 of this chapter counted the frequency of safety interactions between the government, China Railway, social forces, and auxiliary enterprises to create direct relations matrix. The direct relation matrix obtained after aggregation of 224 accident reports is presented in Table 4. And the scores in Table 4 indicate the frequency of inadequate safety-related interactions among the organization nodes and direct adverse effects. It is important to note that different from the traditional method DEMATEL, the generation of the direct relation matrix does not require an expert's opinion. The reason for this conclusion is that influence relationships in this article are extracted from the accident report.

Step2: Normalized matrix N

In the second step, normalize the direct relation matrix using Eq (5). The normalized matrix N is shown in Table 5.

The direct relation diagram shown in Fig 4 is drawn using a normalization matrix, where the arrows indicate the direction of influence of the nodes and the thickness of the lines indicate the intensity of the direct influence between the nodes.

Step3: Total relation matrix T

From the normalized matrix, total relation matrix T was computed using Eq (6) and resulting matrix is shown in Table 6.

There are direct and indirect relations among nodes in the total relation matrix, thus the degree of comprehensive influence among nodes are reflects. Total relationships among nodes can be drawn according to influencing degree and influencing direction, as shown in Fig 5.

Step4: Sum of in degree value $C_i$ and out degree value $D_i$

In the railway safety supervision and management model, each node influences other nodes and is influenced by other nodes simultaneously. As shown in Fig 6, the number of influencing relations is defined as the out-degree, and the number of influenced relations is defined as the in-degree. In this paper, the in and out-degree is three. Finally, the total in and out influencing degree of each dimension of the node is obtained by using Eqs (7) and (8), as shown in Table 7.

Step 5: Designing effect-relationship map

The causal degree is used to judge the node type and is denoted as $R_i$, $R_i = D_i - C_i$. The central degree indicates the degree of influence of the node factor on the stability of the railway safety supervision system, which is recorded $M_i$. $M_i = D_i + C_i$. The calculation results of each index value of the node are shown in Table 7. According to Table 7, central degree and causal

**Table 5. Normalized matrix.**

|  | Government | China Railway | Social force | Auxiliary enterprises |
|---|---|---|---|---|
| Government | 0 | 0.273 | 0.182 | 0.273 |
| China Railway | 0.091 | 0 | 0.091 | 0.182 |
| Social force | 0.091 | 0.545 | 0 | 0.091 |
| Auxiliary enterprises | 0.091 | 0.818 | 0.091 | 0 |

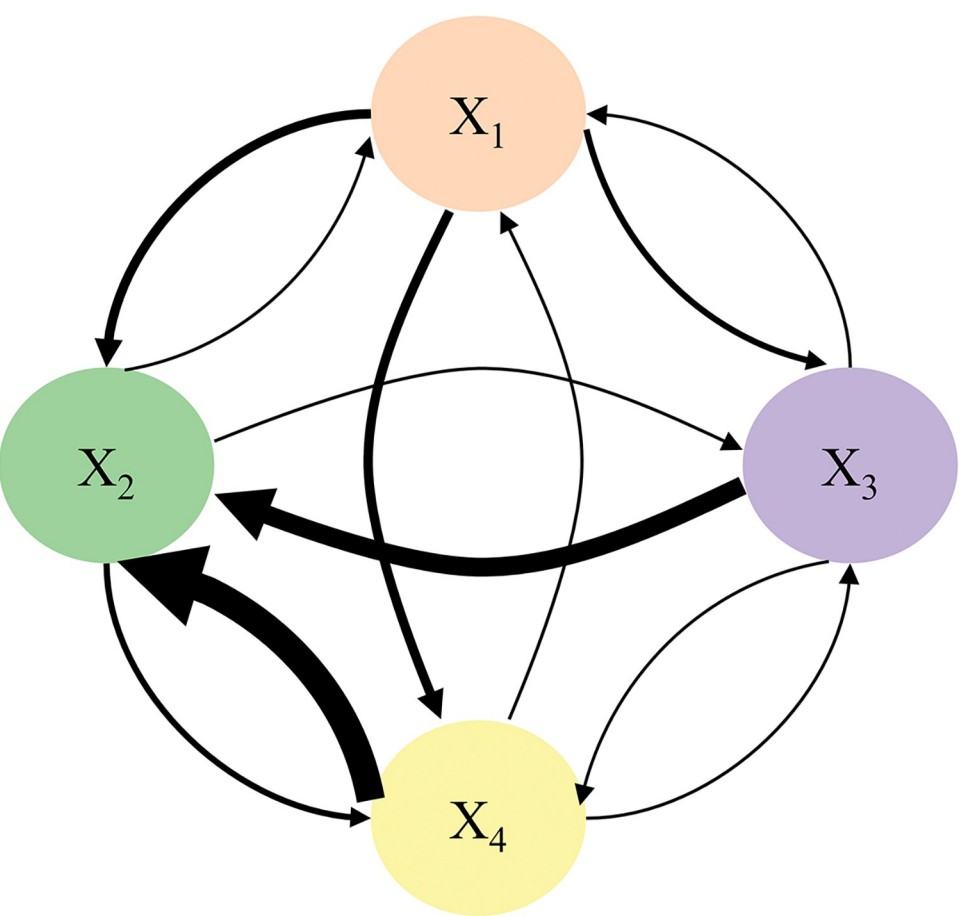

**Fig 4. Diagram of nodes direct influence relationship.**

degree are selected to evaluate the influencing effect of first-level nodes, as shown in Fig 7. The horizontal axis value is the degree of centrality (D+C), and the vertical axis value is the degree of cause (D-C). The interaction effect of nodes in the model determines their position on the axis.

## 4.3 Results

The following relationships can be obtained by comparing the magnitude of support for each organizational node yields:

China Railway (92.793%)> auxiliary enterprises (59.91%)> government (40.991%)> social force (34.234%)

Support indicates the frequency of occurrence of nodes with safety-related interactions in 224 accident reports. It can be seen that China Railway and auxiliary enterprises have the

**Table 6. Total relation matrix.**

|  | Government | China Railway | Social force | Auxiliary enterprises |
|---|---|---|---|---|
| Government | 0.162 | 0.926 | 0.343 | 0. 517 |
| China Railway | 0.175 | 0.415 | 0.190 | 0.322 |
| Social force | 0.226 | 0.077 | 0.161 | 0.345 |
| Auxiliary enterprises | 0.270 | 1.331 | 0.292 | 0.342 |

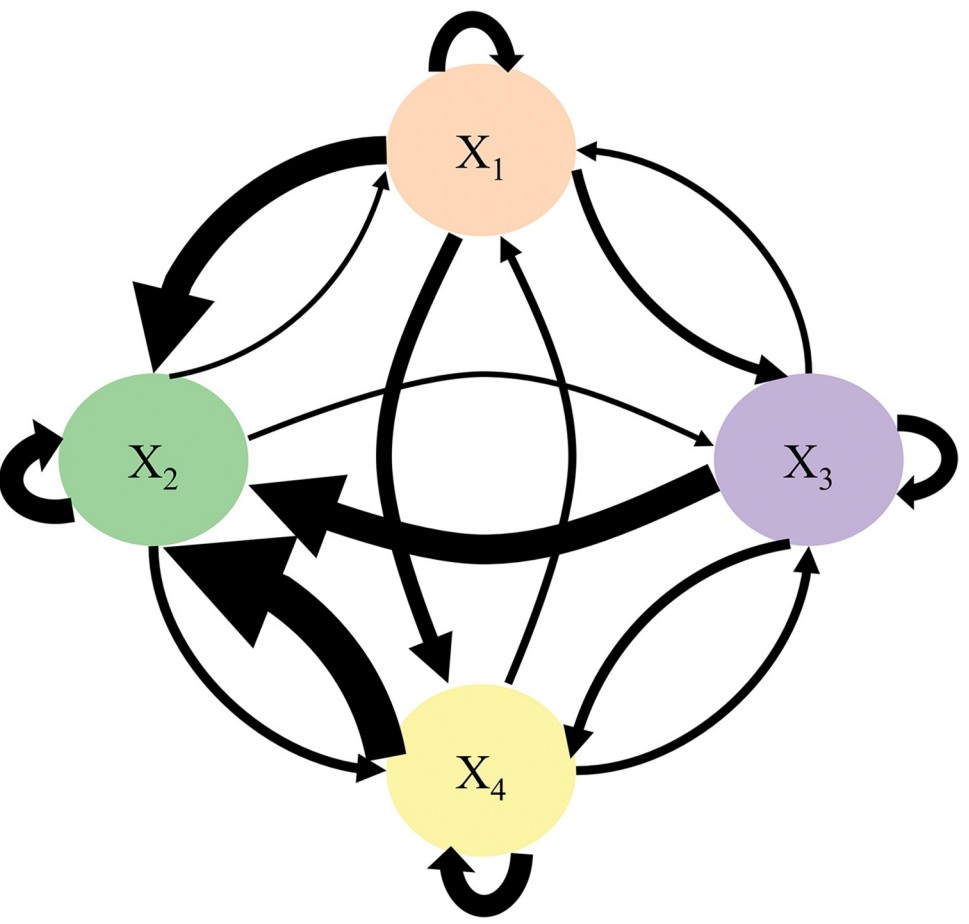

**Fig 5. Diagram of nodes total influence relationship.**

highest frequency, both of which are higher than 50% indicating that they are the primary organizations that are much accounted while conducting railway safety supervision and management. Four association rules for organizations with safety-related interactions are proposed (see Table 3). As can be seen from Table 3, there is the same consequence between rule 3 and rule 2, and the antecedent organizations in rule 3 contain the antecedent organizations in rule 2. Therefore, rule 2 is reserved, because if rule 2 takes effect, rule 3 must take effect according to the nature of the antecedent predicting the consequent in association rule.

Table 3 displays the results of rule 1, where 31.98% of accidents are triggered by unbefitting actions in China Railway node and social force node simultaneously, with a level of confidence of 93.42%. These results demonstrate that those accidents caused by inappropriate behavior in the social force is also linked to inadequate management of China Railway.

The rule 2 including government, social force node and auxiliary enterprises node, and according to rule 2, with 56.25% confidence that those accidents caused by social force and auxiliary enterprises are also accompanied by a lack of government regulation.

The nodes involved in rule 4 are the same as rule 1, therefore, rule 1 and rule 4 have the same support. From the confidence level, the probability of accident caused by China Railway is 34.47%, and it is important to note that social forces are also involved in these accidents. It is apparent from the Figs 4 and 5 that each node's direct and total influence degree are ranked as follows respectively (The degree of influence between absent nodes is equal to the degree of

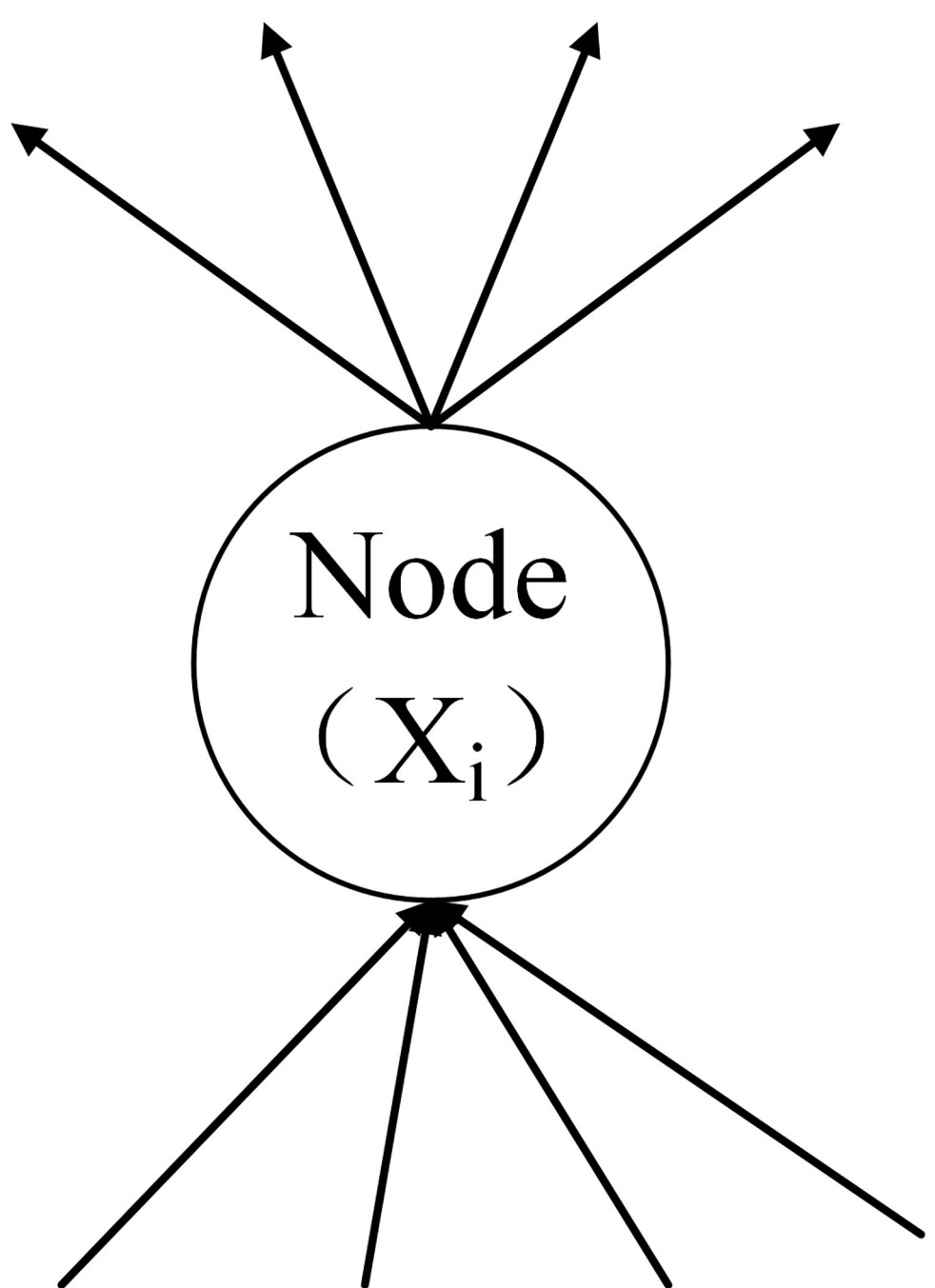

**Fig 6. The diagram of nodes in-out degree.**

Table 7. Prominence and relation results obtained by DEMATEL.

| Node | D | C | D+C | D-C | Weighted value |
|------|------|------|------|------|------|
| $X_1$ | 1.948 | 0.833 | 2.781 | 1.115 | 0.199 |
| $X_2$ | 1.102 | 3.649 | 4.752 | -2.547 | 0.340 |
| $X_3$ | 1.709 | 0.986 | 2.694 | 0.723 | 0.193 |
| $X_4$ | 2.234 | 1.526 | 3.760 | 0.709 | 0.269 |

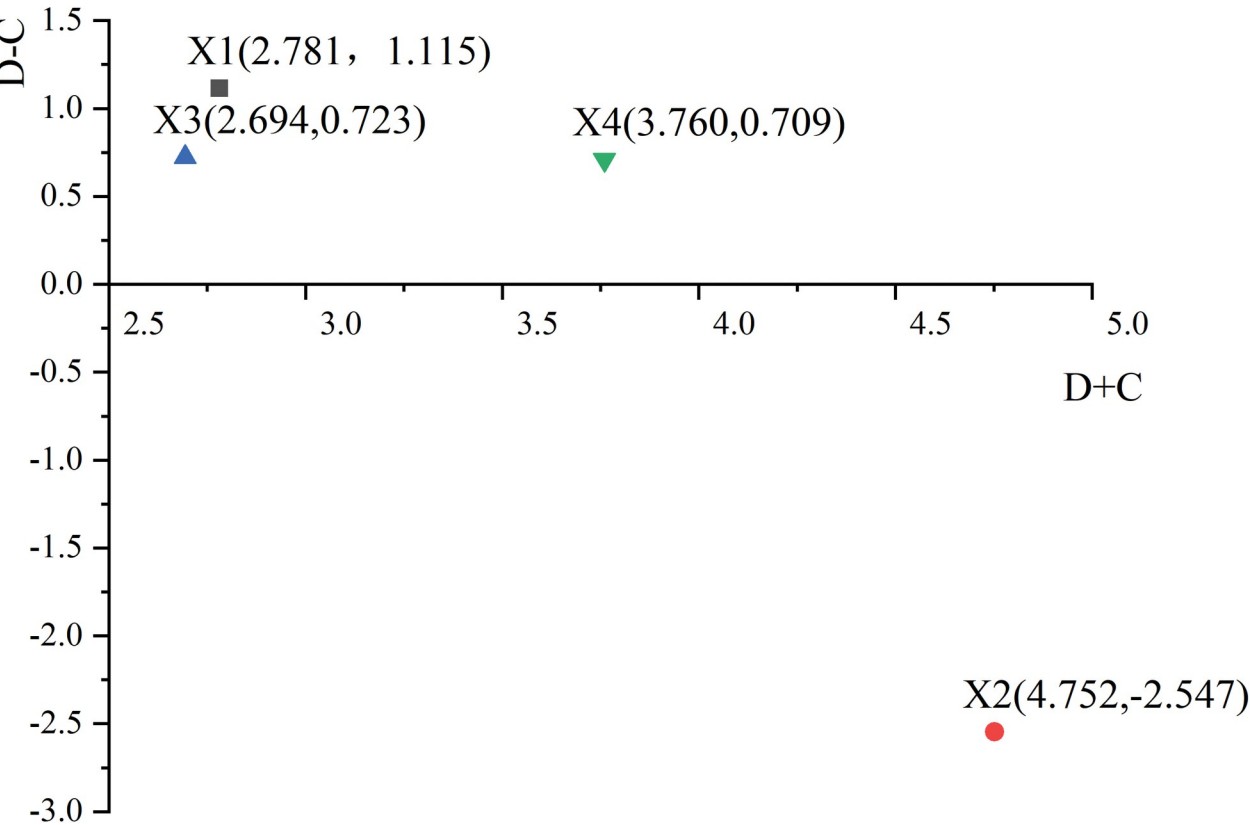

**Fig 7. The cause-effect relationship diagram of nodes.**

influence between $X_2$ nodes and $X_1$ nodes):

$$(X_4, X_2) > (X_3, X_2) > (X_1, X_2) = (X_1, X_4) > (X_1, X_3) = (X_2, X_4) > (X_2, X_1)$$

$$(X_4, X_2) > (X_3, X_2) > (X_1, X_2) > (X_1, X_4) > (X_3, X_4) > (X_1, X_3) > (X_2, X_4) > (X_4, X_3)$$
$$> (X_4, X_1) >$$

$$(X_3, X_1) > (X_2, X_3) > (X_2, X_1)$$

"D+C" denotes the relative weighting of each component in the overall analyzation structure and "D-C" denotes the degree of influence organizational nodes [50]. The prominence of the four organization nodes ranks from the largest to the smallest as follows: $X_2$(China Railway)> $X_4$(auxiliary businesses)> $X_1$(government)> $X_3$(social force). This result is consistent with the support calculation. Besides, out of the four organization nodes, one organization node (i.e., $X_2$: China Railway) is evaluated as recipient, and three organization nodes (i.e., $X_1$: government; $X_3$: social force $X_4$: auxiliary businesses) are evaluated as influencer named cause nodes based on Fig 7. The causal degree of the three cause nodes is sorted as $X_1 > X_3 > X_4$, of which the government node has the highest causal degree, and the social force and auxiliary enterprises have close causal degrees.

**Table 8. The first level nodes control measures.**

| Node | Control measures |
|---|---|
| China Railway | Design phase:<br>(a) Guarantee the status of the main body of the project and do well in supervision;<br>(b) Integrate strategic thinking and realize the diversification of market products;<br>(c) Optimize the production structure through the process re-engineering, department reorganization and virtual operation.<br>Operation phase:<br>(a) Perform its functions and change management concepts and methods;<br>(b) Help Railway administrations with their daily operations;<br>(c) Strictly guard the safety standards and build a safety guarantee system.<br>After accidents:<br>(a) Apply social forces appropriately;<br>(b) Share accident information and handle the emergency;<br>(c) Write an accident report and issue rectification measures after all remedial work. |
| Auxiliary enterprises | (a) The manufacturing industry should control qualities and provide superior products;<br>(b) Safety service enterprises should know the development demands, utilize market mechanism, release the government's pressure, and promote the marketization and industrialization of safety services;<br>(c) High-tech enterprises should keep close to market demands, anticipate trends, and innovate. |
| Government | Design phase:<br>(a) Improve the preliminary system safeguard and supervision;<br>(b) Establish the safety precaution mechanism and contingency plan;<br>(c) During the construction, suspend the work while in problems and continue the work while removing doubts.<br>Operation phase:<br>(a) Focus on the key links of operation and the safety of facilities;<br>(b) Strengthen supervisions;<br>(c) Urge the implementation of enterprises.<br>After accidents:<br>(a) Coordinate the rescue operation;<br>(b) Lead joint law enforcement to investigate and handle the accident after the complement of the emergency. |
| Social forces | (a) Establish a sound management system;<br>(b) Take the initiative to foster participation;<br>(c) News media should take the initiative to transmit information and play the role of cultural preachers;<br>(d) Commonweal organization should strive for charitable resources, build a docking platform, and play an assisting and supplementary role;<br>(e) Interest organization should learn its own ability and make reasonable demands in public decision. |

## 5 Safety measures and proposals

The safety promotion measures proposed in this section comprise two categories. One of them is proposed to promote safety within the nodes and is called control nodes. And the other is proposed to enhance safety interactions among the organizations nodes is called reinforcement measures. According to the rules described above, the relevant safety managers can choose the appropriate measures for railway safety management from Tables 8 and 9.

## 6 Discussions

As a result of the reform of China's railways, China Railway has gained full autonomy. The formation of this situation has led scholars to ignore the safety forces that exist outside the China Railway. Therefore, in order to improve the existing safety supervision and management system of China, auxiliary enterprises and social forces and describes the responsibilities of the organizations are brought in this paper to make up the system. In addition, the model provides a framework for the following study to examine the tools for researching the relationship

**Table 9. The chain reinforcement measures.**

| Influencing node | Influenced node | Reinforcement Measures |
|---|---|---|
| Government | China Railway | Provide a good development environment and institutional innovation. |
| | Social forces | (a) Establish a mass reward system and incorporate it into the coordination mechanism; <br> (b) Improve public opinion collection and guidance methods and advance news legislation. |
| | Auxiliary enterprise | (a) Formulate systems and standards and strengthen supervision; <br> (b) Introduce policies to support development. |
| China Railway | Government | Provide technological innovation. |
| | Social forces | Establish a "communication house", create and improve incentive mechanisms, and encourage the mass to "speak boldly". |
| | Auxiliary enterprise | Strengthen communication. |
| Social forces | Government | (a) Focus on oversight; <br> (b) Help government assistance. |
| | China Railway | Play the role of supervisor, advisor and risk averter. |
| | Social forces | (a) Pay attention to news publicity and strengthen the guiding force of public opinion; <br> (b) Pay attention to vulnerable groups and improve and standardize personnel management system; <br> (c) Give full play to the role of the main body of supervision and strengthen the constraints on power. |
| | Auxiliary enterprise | Strengthen the sense of responsibility and public opinion supervision. |
| Auxiliary enterprise | Government | Strengthen the diversification of service content and the construction of service quality. |
| | China Railway | Continuously inject new technologies and continue to provide services. |
| | Social forces | Take advantage of the positive social influence of social forces. |
| | Auxiliary enterprise | (a) Promote self-discipline of safe behaviors through their rights; <br> (b) Strengthen cooperation self-construction, standardize behavior and provide high quality services. |

between organizations and accidents, as well as the relationship between safety-related interactions among the organizations and accidents. And strategies for avoiding accidents by strengthening organization node and safety-related interactions among the organizations are explored.

For researching the relationship between organizations and accidents, as well as the relationship between safety-related interactions among organizations and accidents, 224 accident reports through association rule and DEMATEL methods are analyzed. It is notable that accidents are closely related to the China Railway node, due to the fact that China Railway node is responsible for both the operation and the supervision and management of the process. The strong correlation between the nodes obtained from Fig 3 shows that the China Railway node is closely connected to all the remaining three nodes, which again reflects the importance of this node in the accident. In addition, the strong correlations suggest that the inadequacy and failure of safety interactions between China Railway and auxiliary enterprises, government and China Railway, China Railway and social force, and government and auxiliary enterprises are also important causes of accidents. As shown in Table 3, the occurrence of the accidents mainly follows as rules: (1) The accident caused by the China Railway is also attributable to the social force. According to this rule, the hazards connected to an occurrence inside the China Railway node should be examined within the social force node based on the nature of the hazard. (2) Accidents caused by social force and auxiliary enterprises nodes are also attributed to government. Considering the supervisory function of the government, this rule can be interpreted to mean that when safety risks are found at the node of social force and auxiliary enterprises, the government should consider the inadequacy of the existing mechanism and system and help the social force and auxiliary enterprises to eliminate the safety risks by strengthening the supervisory strength and other measures. (3) Accidents caused by social force are also attributed to China Railway. Social forces are primarily composed of social organizations and individuals. There is no relatively perfect management system compared with the government or enterprise organizations. Thus, besides eliminating its safety risks, the China Railway should

also rely on its own sound organizational and management system to help eliminate the hidden risks in the social force node.

On the other hand, as shown in Figs 4 and 5, the degree of influence among $(X_4, X_2)$, $(X_3, X_2)$, $(X_1, X_2)$, and $(X_1, X_4)$ comes out on top. The result is more consistent with the strong correlation illustrated in the above paper, which further suggests that insufficient safety interactions and the spread of undesirable effects between the government and auxiliary enterprises, auxiliary enterprises and China Railway, social forces and China Railway, and government and China Railway are the leading causes of accidents. In addition, Fig 5 shows the self-influence of nodes, which arises from the propagation of the influence of the current node. The self-impact ranking of the four nodes is as follows: China Railway> auxiliary enterprises> government≈ social force. The high self-impact may be due to the inability of other system nodes to resist and mitigate the risk, as well as the node's susceptibility to influence.

The results of the DEMATEL study confirm these suspicions. In the term of the value of "D +C" for each organization node, $X_2$(China railway) ranks the first, while the cause degree is negative. Thus, it indicates that China Railway is significantly influenced by other organizations, which is related with the role China Railway played that the main body of railway operation. Therefore, the safety measures formulated should promote the stability of China Railway nodes, which can strengthen the ability of China Railway to resist external risks. The government has the largest "D-C" value, and the value of "D" ranks second. It suggests that government intervention plays more influence on other nodes. The reason may be that the government is the most effective regulator, which constrains the behavior of other nodes by promulgating laws and regulations. In addition to government node, the other two cause nodes are Auxiliary enterprises and social force. Auxiliary enterprises have a maximum D value. In addition, it has the largest D+C value compared to the other three cause nodes. Considering the close relationship between auxiliary enterprises and China Railway in practice. It suggests that auxiliary enterprises play a great influence on China Railway compared to the remaining two cause nodes. Social force, which has lacked attention in previous research, can be relied upon as a cause node to promote railway safety through features such as high flexibility.

## 7 Conclusions

The main contribution of this study is a safety supervision and management model consisting of organizations that are relevant for cooperative safety management. Need of special note is that this paper innovates to add social forces and auxiliary enterprises as components of railway safety management. The model also provides an analytical framework for studying the relationship between organization and accidents, as well as safety interaction and accidents.

The identification of the relationship between organizations and accidents, as well as the relationship between interactions among the organizations and accidents help railway experts and engineers to prevent the accident caused by organizations, and to promote railway transport safety. As a consequence, the study concludes that China Railway is the most essential organization that causes accidents and is also the effect node that is susceptible to the adverse influence of other organizations. government, social force, and auxiliary enterprises are cause nodes that behavior of these nodes can easily affect the China Railway. Moreover, four rules that should be considered for preventing accidents are: (1) Prevent the occurrence of accidents at the China Railway node and the social force node simultaneously; (2) The government should act promptly to remove potential safety dangers when risks are identified simultaneously by both auxiliary enterprises and the social force. (3) Strengthen the safety interaction between the government and China Railway, social force and China Railway, and auxiliary

enterprises and China Railway. (4) Improve the ability of China Railway nodes to resist external risks. Thus, the primary safety promotion measures of this study state as follows:

Twenty-five node safety promotion measures have been proposed. Among them, nine measures and suggestions are proposed to improve supervision and coordination capabilities and strengthen the main functions of railways in three stages. Five measures and suggestions for the social forces node that cultivate and improve the enthusiasm of social forces in participating in railway safety management. Three measures and suggestions are proposed for the auxiliary enterprise node that conducive to railway safety management institutions to realize the development needs, improve the innovation of railway safety-related equipment, and strengthen equipment quality control.

In view of the current situation of insufficient safety interaction between the nodes, 20 measures are proposed to promote the construction of railway safety. Among them, Five suggestions are proposed between the government and China Railway, as well as social forces and auxiliary enterprises to enhance the government's provision of institutional guarantees and reinforce the government's guidance mechanism; Three suggestions are proposed between China Railway and the government, as well as social forces and auxiliary enterprises that to strengthen the communication and promote the development of railway safety assurance technology; Four suggestions between social forces and government, as well as China Railway and auxiliary enterprises that to promote the development of social supervision and assistance work; Five measures are proposed from the perspective of offering technology, equipment, and services to reinforce the relationship between auxiliary enterprises and the government, as well as the relationship between China Railway and social forces.

## Author Contributions

**Conceptualization:** Jia Liu, Yansheng Wang, Cunbao Deng, Zhixin Jin, Gaolei Wang, Chen Yang, Xiaoyu Li.

**Data curation:** Jia Liu, Gaolei Wang.

**Methodology:** Jia Liu, Yansheng Wang, Cunbao Deng, Zhixin Jin, Chen Yang, Xiaoyu Li.

**Supervision:** Gaolei Wang.

**Writing – original draft:** Jia Liu.

**Writing – review & editing:** Yansheng Wang, Cunbao Deng.

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
