## [Decision Letter · Decision Letter 0]

24 Sep 2023

PONE-D-23-27933Research on China railway safety supervision and management system based on DEMATEL and association rulePLOS ONE

Dear Dr. Yansheng,

Thank you for submitting your manuscript to PLOS ONE. After careful consideration, we feel that it has merit but does not fully meet PLOS ONE’s publication criteria as it currently stands. Therefore, we invite you to submit a revised version of the manuscript that addresses the points raised during the review process.

It is important to address the following comments in order to enhance the quality of your paper:

[1] English language: It is important to carefully revise and improve the language throughout the manuscript.

[2] Abstract: The abstract needs to be carefully revised. 

[3] Introduction: The introduction section should be improved. 

[4] Discussion: The discussion and content of the paper are not sufficient. It is recommended to provide more detailed explanations and analysis.

[5] Methodology: The methodology section is considered incomplete and unclear. It is important to revise and provide more clarity to avoid confusion for the readers.

[6] Application of results: Please clarify the application of the paper's results in both the Abstract and Conclusions sections. Conclusion section should be carefully revised.Please submit your revised manuscript by Nov 08 2023 11:59PM. If you will need more time than this to complete your revisions, please reply to this message or contact the journal office at plosone@plos.org. Please include the following items when submitting your revised manuscript:A rebuttal letter that responds to each point raised by the academic editor and reviewer(s). You should upload this letter as a separate file labeled 'Response to Reviewers'.A marked-up copy of your manuscript that highlights changes made to the original version. You should upload this as a separate file labeled 'Revised Manuscript with Track Changes'.An unmarked version of your revised paper without tracked changes. You should upload this as a separate file labeled 'Manuscript'.

We look forward to receiving your revised manuscript.

Kind regards,

Ibrahim Badi, PhD

Academic Editor

PLOS ONE

Journal Requirements:

"The authors would like to appreciate editors and reviewers for their constructive comments and suggestions. The research described in this paper was financially supported by National Natural science Foundation of China (52004175) and Science and technology innovation project of colleges and universities in Shanxi Province (2020L0105). The authors would like also to thanks for all the supports to publish this paper."

"The research described in this paper was financially supported by National Natural science Foundation of China (52004175) and Science and technology innovation project of colleges and universities in Shanxi Province (2020L0105) from Yansheng Wang. And Yansheng Wang play the roles in the study design and preparation of the manuscript."

Reviewers' comments:

Reviewer's Responses to Questions

**Comments to the Author**

1. Is the manuscript technically sound, and do the data support the conclusions?

Reviewer #1: Yes

2. Has the statistical analysis been performed appropriately and rigorously? 

Reviewer #1: Yes

3. Have the authors made all data underlying the findings in their manuscript fully available?

Reviewer #1: Yes

4. Is the manuscript presented in an intelligible fashion and written in standard English?

Reviewer #1: No

5. Review Comments to the Author

Reviewer #1: 1. English is not good and coherent.

2. The abstract needs to be revised. Starting with the limitation of previous studies and novelty of the paper.

3. Introduction is so weak and needs to be revised using previous papers. The authors are strongly recommended to cite all these papers and those like them:

- Sadeghi, J., & Essmayil Kaboli, M. (2015). Investigation of the influences of track superstructure parameters on ballasted railway track design. Civil Engineering Infrastructures Journal, 48(1), 157-174.

- Hasheminezhad, A., Hadadi, F. & Shirmohammadi, H. Investigation and prioritization of risk factors in the collision of two passenger trains based on fuzzy COPRAS and fuzzy DEMATEL methods. Soft Comput (2021). https://doi.org/10.1007/s00500-020-05478-3

4. There are some grammatical errors through the paper needs to be corrected.

5. There are many figures and tables in the paper but the discussion and content of the paper is not enough.

6. The methodology section is not completed and clears enough and will make many questions in the readers mind.

7. What is the application of the paper results? It must be clarified in the sections Abstract and Conclusions.

8. The section Conclusion is not written well. It must show the main findings of the paper.

6. PLOS authors have the option to publish the peer review history of their article (what does this mean?). If published, this will include your full peer review and any attached files.

Reviewer #1: No

---

## [Author Response · Author response to Decision Letter 0]

7 Nov 2023

Dear Editors and Reviewers:

Thanks for you and the reviewers’ comments concerning our manuscript entitled “Research on China railway safety supervision and management system based on DEMATEL and association rule” (manuscript number is PONE-D-23-27933). Those comments are all valuable and very helpful for revising and improving our paper, as well as the important guiding significance to our researches. We have checked the manuscript and revised it according to the comments. The main corrections in the paper and the responds to the reviewers’ comments are as flowing:

Reviewer #1:

Comment 1: “English is not good and coherent.” 

Reply: We are very sorry for incoherent English. We tried our best to improve the manuscript and made some changes to the manuscript. These changes will not influence the content and framework of the paper. And here we did not list the changes but marked in red in the revised paper. We appreciate for Editors/Reviewers’ warm work earnestly and hope that the correction will meet with approval.

Comment 2: “The abstract needs to be revised. Starting with the limitation of previous studies and novelty of the paper.” 

Reply: Thank you very much for your reminder, we are very sorry that we wrote the abstract not good enough. We have rewritten the abstract that start with the limitation of previous studies and novelty of the paper. The revised abstract in the revised manuscript have been marked in red. The revised abstract is as follows:

Safety management is a key issue in the railroad industry that needs to be continuously focused on. And it is essential to study causes of accidents for preventing accidents. However, there is a limited academic discussion on the systematic study of organizations and accidents, as well as their safety-related interactions and accidents, as opposed to human-caused disasters and defects in equipment and technology. Thus, the model of China railway safety supervision and management system by sorting out the existing organizations involved in management in China is established in this paper firstly, social forces and auxiliary enterprises are specifically added to the model. And then, the relationship between organizations and accidents, as well as the relationship between safety interactions among organizations and accidents are explored by analyzing 224 accident reports, which led to 4 principles for accident prevention. Finally, based on these principles, measures to secure organizational nodes, as well as measures to promote safe interactions among organizations are proposed. The results showed that: China railway node is not only the most critical node in the safety supervision and management system but also the most vulnerable to the influence of other nodes. (2) The accident occurred due to the simultaneous occurrence of an accident at the China railway node and the social force node. (3) When there are often safety risks in auxiliary enterprises and social forces simultaneously, the government's management is likely to be defective. The findings in this study can provide helpful references not only for improvement of safety management system structure and supervision and management mechanism but also for the formulation of safety supervision and management policies in China and other countries.

Comment 3: “Introduction is so weak and needs to be revised using previous papers. The authors are strongly recommended to cite all these papers and those like them:

- Sadeghi, J., & Essmayil Kaboli, M. (2015). Investigation of the influences of track superstructure parameters on ballasted railway track design. Civil Engineering Infrastructures Journal, 48(1), 157-174.

- Hasheminezhad, A., Hadadi, F. & Shirmohammadi, H. Investigation and prioritization of risk factors in the collision of two passenger trains based on fuzzy COPRAS and fuzzy DEMATEL methods. Soft Comput (2021). https://doi.org/10.1007/s00500-020-05478-3.”

Reply: Thank you very much for your reminder, we are very sorry that we wrote the introduction too weak. We have rewritten the Introduction and added the references that include your recommendation. The revised Introduction and relevant references in the revised manuscript have been marked in red. The revised abstract and references are provided below for your quick reference.

Rail is becoming a more critical mode of transportation in China which promotes the development of raising the standard of living of the people[1, 2]. However, with the continuous development and construction of railway transportation, railway accidents occur frequently and bring serious consequences, including casualties, property damage, and poor social impact, which greatly threatens the further development of the railway industry[3]. The guarantee of railway safety depends on the progress of related technology, the improvement of organization and system, and effective safety management.[4, 5]. For the rail, academic and practical attention has focused mostly on the role of technical and human-caused disasters such as improper structural design of rolling stock and misfeasance of human [6-8]. For example, the Wenzhou train crash accident on July 23, 2011, and with serious consequences. The accident was caused by an equipment defect[9]. However, accidents have not been caused by a coincidence of independent failures such as defect of equipment, but by a systematic migration of organizational behavior toward accident, which is described by Rasmussen and Baysari et al. in their article[10, 11]. Previous studies show that 30%-40% of accidents can be attributed to organizational factors and almost all accidents were related to at least one organizational factor in large-scale complex systems. And the reasons are not only related to individual's fault but related to a partial or total failure of the organization[10, 12, 13]. The interaction problem of the organization may be another reason that affects safety[14, 15]. Synergies among organizations that emerge from frequent interaction and communication between organizations are considered to be the basis for achieving system functionality[16]. Deficiencies in reliability of synergy among organizations lead to unexpected situations where risks are increase and rail can’t be supervised and managed as administrator anticipated[17]. In summary, poor organization structure and insufficient interaction among the organizations can be a significant cause of accidents, which are also related to the management and regulatory[18, 19]. 

It is obvious that government regulation is an inadequate way to enforce safety. Safety-related coordination between the operator and the manufacture, as well as the coordination between media and regulators are considered to be effective ways to promote safety management[20, 21]. Thus, organizations should take on the responsibility to respond to risks[22]. Organizations must adapt their structure to follow the changing safety objectives[23]. In China, the organizational structure of the railway industry changed in 2013 to a government-regulated, company-operated situation that continues to this day[24, 25]. Within this organizational structure, the operator-regulator relationship has been identified as crucial causal factor for accident. Meanwhile, the information flow between corporations and the government can be an indication of overall safety[26, 27]. The accident and mortality rate have dropped benefits from change of China. However, a number of accidents caused by inadequate organizations in recent years have drawn the attention of safety personnel. For example, trains T179 and D2809 derailed respectively on March 2020 and June 2022. Moreover, the Baiyin derailment accident that caused serious consequences happened on March 8, 2022 [28, 29]. The accident highlighted the need for further improvement in the current organizational framework of railway safety management. Rail safety is affected by organizational behavior in a variety of ways[30]. Accident analysis methods, such as Model of Socio-Technical[31] and STAMP[32, 33], are seen as an effective tool to help to research =the impact of organizational behavior. Such as Xing[34] provided an administrative structure of accident by using STAMP, and some relationships between organizations and accidents are drawn. However, as this conclusion is based on the analysis of one accident and the complexity of the analysis method, there is a doubt that the conclusion apply to all accidents[35]. Current research elucidates that the process of accidents involves the entire safety management system and highlights the significance of organizations and safety-related interactions among the organizations in this process. Additionally, it reflects the inadequacy of current research: There is a lack of systematic research on accident and organization, as well as accident and safety-related interactions among organizations, especially in the domain of railway industry[36].

Therefore, establishing a safety supervision and management system which applies to China railway and investigating the mechanisms of safety-related interactions among organizations lead to the reduction of accidents and poor impact in the future. In view of this, we established China’s railway safety supervision and management system model and then used DEMATEL and association rule method to research safety-related interactions relationships between organizations in the model[27, 37, 38]. Finally, based on the research results, we proposed some specific measures to strengthen the security of railway management. The above issue has provided a reference for the safe operation of the railway and motivated researchers and engineers in the safety fields of transport to study more reasonable supervision and management operation mechanisms in the railway industry.

1. Cao Y, An Y, Su S, Xie G, Sun Y. A statistical study of railway safety in China and Japan 1990–2020. Accident Analysis & Prevention. 2022;175. doi: 10.1016/j.aap.2022.106764.

2. Tian N, Tang S, Che A, Wu P. Measuring regional transport sustainability using super-efficiency SBM-DEA with weighting preference. Journal of Cleaner Production. 2020;242. doi: 10.1016/j.jclepro.2019.118474.

3. Wang N, Yang X, Chen J, Wang H, Wu J. Hazards correlation analysis of railway accidents: A real-world case study based on the decade-long UK railway accident data. Safety Science. 2023;166. doi: 10.1016/j.ssci.2023.106238.

4. Hasheminezhad A, Hadadi F, Shirmohammadi H. Investigation and prioritization of risk factors in the collision of two passenger trains based on fuzzy COPRAS and fuzzy DEMATEL methods. Soft Comput. 2021;25(6):4677-97. doi: 10.1007/s00500-020-05478-3.

5. Hsu SH, Lee CC, Wu MC, Takano K. The influence of organizational factors on safety in Taiwanese high-risk industries. Journal of Loss Prevention in the Process Industries. 2010;23(5):646-53. doi: 10.1016/j.jlp.2010.06.018. 

6. Burdzik R, Nowak B, Rozmus J, Słowiński P, Pankiewicz J. Safety in the railway industry. Archives of Transport. 2017;44(4):15-24. doi: 10.5604/01.3001.0010.6158.

7. Janota A, Pirník R, Zdánsky J, Nagy P. Human Factor Analysis of the Railway Traffic Operators. Machines. 2022;10(9):820. doi: ARTN 82010.3390/machines10090820. PubMed PMID: WOS:000856890200001.

8. Sadeghi J, Hasheminezhad A, Essmayil Kaboli M. Investigation of the influences of track superstructure parameters on ballasted railway track design. 107508/CEIJ201501011. 2015;48(1):157-74.

9. Zhan Q, Zheng W, Zhao B. A hybrid human and organizational analysis method for railway accidents based on HFACS-Railway Accidents (HFACS-RAs). Safety Science. 2017;91:232-50. doi: 10.1016/j.ssci.2016.08.017.

10. Baysari MT, McIntosh AS, Wilson JR. Understanding the human factors contribution to railway accidents and incidents in Australia. Accident Analysis & Prevention. 2008;40(5):1750-7. doi: 10.1016/j.aap.2008.06.013.

11. Rasmussen J. Risk management in a dynamic society: a modelling problem. Safety Science. 1997;27(2-3):183-213. doi: 10.1016/s0925-7535(97)00052-0.

12. Hsu SH, Lee C-C, Wu M-C, Takano K. The influence of organizational factors on safety in Taiwanese high-risk industries. Journal of Loss Prevention in the Process Industries. 2010;23(5):646-53. doi: 10.1016/j.jlp.2010.06.018.

13. Hollnagel E, Woods DD. Joint Cognitive Systems: CRC press; 2005.

14. Mochalin SM, Tyukina LV, Novikova TV, Pogulyaeva IV, Romanenko EV. Problems of Inter-organizational Interaction of Participants in Motor Transport Cargo Shipments. Indian Journal of Science and Technology. 2016;9(21):95220. doi: 10.17485/ijst/2016/v9i21/95220.

15. Stroeve SH, Sharpanskykh A, Kirwan B. Agent-based organizational modelling for analysis of safety culture at an air navigation service provider. Reliability Engineering & System Safety. 2011;96(5):515-33. doi: 10.1016/j.ress.2010.12.017.

16. Pezzillo Iacono M, Schiuma G, Martinez M, Mangia G, Galdiero C. Knowledge creation and inter‐organizational relationships: the development of innovation in the railway industry. Journal of Knowledge Management. 2012;16(4):604-16. doi: 10.1108/13673271211246176.

17. Pöllänen M, Liimatainen H. Synergies and conflicts between safety and environmental measures in transport. 2014.

18. Li W-C, Harris D, Yu C-S. Routes to failure: Analysis of 41 civil aviation accidents from the Republic of China using the human factors analysis and classification system. Accident Analysis & Prevention. 2008;40(2):426-34. doi: 10.1016/j.aap.2007.07.011.

19. Deng Y, Hu Q, Tan M, Lu H, Zhu Y. A rough set-based measurement model study on high-speed railway safety operation. Plos One. 2018;13(6). doi: 10.1371/journal.pone.0197918.

20. Ferjencik M. Totalitarian loss of responsibility in an explosives production plant. Safety Science. 2011;49(2):253-67. doi: 10.1016/j.ssci.2010.08.006.

21. Kawakami S. Application of a systems-theoretic approach to risk analysis of high-speed rail project management in the US. Massachusetts Institute of Technology. 2014.

22. Fan Y, Li Z, Pei J, Li H, Sun J. Applying systems thinking approach to accident analysis in China: Case study of “7.23” Yong-Tai-Wen High-Speed train accident. Safety Science. 2015;76:190-201. doi: 10.1016/j.ssci.2015.02.017.

23. Kontogiannis T. A contemporary view of organizational safety: variability and interactions of organizational processes. Cogn Technol Work. 2010;12(4):231-49. doi: 10.1007/s10111-009-0131-x.

24. Huang W, Zhang Y, Shuai B, Xu M, Xiao W, Zhang R, et al. China railway industry reform evolution approach: Based on the Vertical Separation Model. Transportation Research Part A: Policy and Practice. 2019;130:546-56. doi: 10.1016/j.tra.2019.09.049.

25. Yu H. Railway Sector Reform in China: controversy and problems. Journal of Contemporary China. 2015;24(96):1070-91. doi: 10.1080/10670564.2015.1030957.

26. Bugalia N, Maemura Y, Ozawa K. Organizational and institutional factors affecting high-speed rail safety in Japan. Safety Science. 2020;128. doi: 10.1016/j.ssci.2020.104762.

27. Caird JK, Kline TJ. The relationships between organizational and individual variables to on-the-job driver accidents and accident-free kilometres. Ergonomics. 2004;47(15):1598-613. Epub 2004/11/17. doi: 10.1080/00140130412331293355. 

28. Guo M, Liu S, Chu F, Ye L, Zhang Q. Supervisory and coworker support for safety: Buffers between job insecurity and safety performance of high-speed railway drivers in China. Safety Science. 2019;117:290-8. doi: 10.1016/j.ssci.2019.04.017.

29. Li KH, Zhang YD, Guo J, Ge XC, Su YB. System dynamics model for high-speed railway operation safety supervision system based on evolutionary game theory. Concurr Comp-Pract E. 2019;31(10):e4743. doi: https://doi.org/10.1002/cpe.4743.

30. Sadeghi J, Hasheminezhad A. Correlation between rolling noise generation and rail roughness of tangent tracks and curves in time and frequency domains. Applied Acoustics. 2016;107:10-8. doi: 10.1016/j.apacoust.2016.01.006.

31. Rasmussen J, Suedung I. Proactive risk management in a dynamic society: Swedish Rescue Services Agency; 2000.

32. Leveson N. A new accident model for engineering safer systems. Safety Science. 2004;42(4):237-70. doi: 10.1016/s0925-7535(03)00047-x.

33. Zhang Y, Dong C, Guo W, Dai J, Zhao Z. Systems theoretic accident model and process (STAMP): A literature review. Safety Science. 2022;152. doi: 10.1016/j.ssci.2021.105596.

34. Xing JD, Meng HX, Meng XK. An urban pipeline accident model based on system engineering and game theory. Journal of Loss Prevention in the Process Industries. 2020;64:104062. doi: ARTN 10406210.1016/j.jlp.2020.104062.

35. Salmon PM, Cornelissen M, Trotter MJ. Systems-based accident analysis methods: A comparison of Accimap, HFACS, and STAMP. Safety Science. 2012;50(4):1158-70. doi: 10.1016/j.ssci.2011.11.009.

36. Milch V, Laumann K. Interorganizational complexity and organizational accident risk: A literature review. Safety Science. 2016;82:9-17. doi: 10.1016/j.ssci.2015.08.010.

37. Agrawal R, Srikant R, editors. Fast Algorithms for Mining Association Rules in Large Databases. Very Large Data Bases Conference; 1994.

38. Si S-L, You X-Y, Liu H-C, Zhang P. DEMATEL Technique: A Systematic Review of the State-of-the-Art Literature on Methodologies and Applications. Mathematical Problems in Engineering. 2018;2018:1-33. doi: 10.1155/2018/3696457.

Comment 4: “There are some grammatical errors through the paper needs to be corrected” 

Reply: We are very grateful to Reviewer for reviewing the paper so carefully. We have tried our best to improve the grammar of manuscript and have modified some confusing sentences, making them concise and easy to read.

Comment 5: “There are many figures and tables in the paper but the discussion and content of the paper is not enough.” 

Reply: Thank you very much for your reminder. We have enriched the description and discussion of figures and tables. The revised in the revised manuscript have been marked in red.

Comment 6: “The methodology section is not completed and clears enough and will make many questions in the readers mind.” 

Reply: We are very sorry that we provided a confusing content in methodology section. We reorganized the structure of the article in the revised manuscript. The methodology has been refined in Section 3: Materials and methods. The new structure is:

3. Materials and methods

3.1 Data preparation

3.2 Association rule and DEMATEL method

3.2.1 Association rule method

3.2.2 Dematel method

4. Analysis and results

4.1 Association rule mining of the relationship between nodes and accidents

4.2 Determination of the interrelationships between nodes

4.3 Results

Comment 7: “What is the application of the paper results? It must be clarified in the sections Abstract and Conclusions.” 

Reply: Thank you very much for your reminder, we are very sorry that we did ignore the important role of the application of the paper results in Abstract and Conclusions. We carefully reviewed the manuscript. The supplementary parts have been marked in yellow in Abstract and Conclusions.

Comment 8: “The section Conclusion is not written well. It must show the main findings of the paper.” 

Reply: Thank you very much for your reminder, we are very sorry that we did not show the main findings in Conclusion. We have rewritten the Conclusion that show the main findings of the paper. The revised Conclusion in the revised manuscript have been marked in red. The revised Conclusion is as follows:

The main contribution of this study is a safety supervision and management model consisting of organizations that are relevant for cooperative safety management. Need of special note is that this paper innovates to add social forces and auxiliary enterprises as components of railway safety management. The model also provides an analytical framework for studying the relationship between organization and accidents, as well as safety interaction and accidents.

The identification of the relationship between organizations and accidents, as well as the relationship between interactions among the organizations and accidents help railway experts and engineers to prevent the accident caused by organizations, and to promote railway transport safety. As a consequence, the study concludes that China Railway is the most essential organization that causes accidents and is also the effect node that is susceptible to the adverse influence of other organizations. government, social force, and auxiliary enterprises are cause nodes that behavior of these nodes can easily affect the China Railway. Moreover, four rules that should be considered for preventing accidents are: (1) Prevent the occurrence of accidents at the China Railway node and the social force node simultaneously; (2) The government should act promptly to remove potential safety dangers when risks are identified simultaneously by both auxiliary enterprises and the social force. (3) Strengthen the safety interaction between the government and China Railway, social force and China Railway, and auxiliary enterprises and China Railway. (4) Improve the ability of China Railway nodes to resist external risks. Thus, the primary safety promotion measures of this study state as follows:

Twenty-five node safety promotion measures have been proposed. Among them, nine measures and suggestions are proposed to improve supervision and coordination capabilities and strengthen the main functions of railways in three stages. Five measures and suggestions for the social forces node that cultivate and improve the enthusiasm of social forces in participating in railway safety management. Three measures and suggestions are proposed for the auxiliary enterprise node that conducive to railway safety management institutions to realize the development needs, improve the innovation of railway safety-related equipment, and strengthen equipment quality control. 

In view of the current situation of insufficient safety interaction between the nodes, 20 measures are proposed to promote the construction of railway safety. Among them, Five suggestions are proposed between the government and China Railway, as well as social forces and auxiliary enterprises to enhance the government's provision of institutional guarantees and reinforce the government's guidance mechanism; Three suggestions are proposed between China Railway and the government, as well as social forces and auxiliary enterprises that to strengthen the communication and promote the development of railway safety assurance technology; Four suggestions between social forces and government, as well as China Railway and auxiliary enterprises that to promote the development of social supervision and assistance work; Five measures are proposed from the perspective of offering technology, equipment, and services to reinforce the relationship between auxiliary enterprises and the government, as well as the relationship between China Railway and social forces.

Special thanks to you for your good comments.

Other changes:

Change 1: We have rewritten the Discussions. More detailed explanations and analysis have provided in Discussions. The revised Discussions in the revised manuscript have been marked in red.

Change 2: We have rewritten the Title to adapt to the changing structure of this paper. The revised Title in the revised manuscript have been marked in red.

Change 3: We have removed the Acknowledgements in this paper according the Journal Requirements.

We tried our best to improve the manuscript and made some changes in the manuscript. These changes will not influence the content and framework of the paper.

We appreciate for Editors’ and Reviewers’ warm work earnestly, and hope that the correction will meet with approval.

Once again, thank you very much for your comments and suggestions.

Looking forward to hearing from you.

With best regards!

Yours sincerely,

Yansheng Wang

---

## [Decision Letter · Decision Letter 1]

29 Nov 2023

Research on safety supervision and management system of China railway based on association rule and DEMATEL

PONE-D-23-27933R1

Dear Dr. Yansheng,

We’re pleased to inform you that your manuscript has been judged scientifically suitable for publication and will be formally accepted for publication once it meets all outstanding technical requirements.

Kind regards,

Ibrahim Badi, PhD

Academic Editor

PLOS ONE

Additional Editor Comments (optional):

Reviewers' comments:

Reviewer's Responses to Questions

**Comments to the Author**

1. If the authors have adequately addressed your comments raised in a previous round of review and you feel that this manuscript is now acceptable for publication, you may indicate that here to bypass the “Comments to the Author” section, enter your conflict of interest statement in the “Confidential to Editor” section, and submit your "Accept" recommendation.

Reviewer #1: All comments have been addressed

2. Is the manuscript technically sound, and do the data support the conclusions?

Reviewer #1: Yes

3. Has the statistical analysis been performed appropriately and rigorously? 

Reviewer #1: Yes

4. Have the authors made all data underlying the findings in their manuscript fully available?

Reviewer #1: Yes

5. Is the manuscript presented in an intelligible fashion and written in standard English?

Reviewer #1: Yes

6. Review Comments to the Author

Reviewer #1: (No Response)

7. PLOS authors have the option to publish the peer review history of their article (what does this mean?). If published, this will include your full peer review and any attached files.

Reviewer #1: No

---

## [Editor Report · Acceptance letter]

4 Dec 2023

PONE-D-23-27933R1 

Research on safety supervision and management system of China railway based on association rule and DEMATEL 

Dear Dr. Wang:

I'm pleased to inform you that your manuscript has been deemed suitable for publication in PLOS ONE. Congratulations! Your manuscript is now with our production department. 

Kind regards, 

on behalf of

Dr. Ibrahim Badi 

Academic Editor

PLOS ONE